# Wavelet Decomposition in Analysis of Impact of Virtual Reality Head Mounted Display Systems on Postural Stability

**DOI:** 10.3390/s20247138

**Published:** 2020-12-12

**Authors:** Piotr Wodarski, Jacek Jurkojć, Marek Gzik

**Affiliations:** Department of Biomechatronics, Faculty of Biomedical Engineering, Silesian University of Technology, 41-800 Zabrze, Poland; jacek.jurkojc@polsl.pl (J.J.); marek.gzik@polsl.pl (M.G.)

**Keywords:** body stability, stabilography, wavelet decomposition, head mounted display, virtual reality

## Abstract

This study investigated how spatial projection systems influences body balance including postural stability. Analyzing precisely defined frequency bands of movements of the center of pressure makes it possible to determine the effectiveness of the balance system’s response to disruptions and disorders and may be used as an indicator in the diagnosis of motor dysfunction. The study involved 28 participants for whom the center of pressure was assessed in a test with open eyes, closed eyes and with virtual reality projection. Percent distributions of energy during wavelet decomposition were calculated. Changes in body stability were determined for the virtual reality tests and these changes were classified as an intermediate value between the open-eyes test and the closed-eyes test. The results indicate the importance of using safety support systems in therapies involving Virtual Reality. The results also show the necessity of measurements times in stabilographic evaluations in order to conduct a more thorough analysis of very low frequencies of the center of pressure signal.

## 1. Introduction

Mechanisms of balance in humans are comprised of many elements which are linked through complex feedback loops [1,2]. The system aims to prevent both internal and external disruptions of balance and the prevention of falls [3,4]. Maintaining the balance system in good working condition requires regular exercise and training, particularly in the case of elderly people, in the context of an aging society [5,6]. Physical exercises are becoming more and more popular and desirable thanks to development of modern techniques including the use of virtual reality technology. Virtual reality technology increases the attractiveness of physical training and provides feedback in real time, which can positively influence the therapy results [7,8].

The application of virtual reality projection may require that the patient should wear 3D (with three-dimensional projection) goggles or a full Head Mounted Display (HMD) system. This display system is mounted on their head applying additional mass to the head and restricting, partially or fully, the patient’s view, affecting their stability. The impact of HMD systems on stabilographic values such as range of postural sways of the center of pressure (CoP), mean velocity of the CoP and maximum velocity of the CoP have been reported in the research conducted by Robert et al. [9].

Assessing the impact of virtual reality on changes in postural stability can be conducted similar to investigations done by Robert et al. [9] using classic tests based on evaluating displacements of CoP within a specified time interval. This test is often performed during a 30 s Romberg’s test in which measurements are conducted in standing position, motionless posture with open and closed eyes and results are compared with each other [9]. On the basis of the obtained displacements, classic stabilographic values are calculated which include path or ellipse surface area. However, these values do not provide information on how the patient was balancing their body during the test. Determination of changes in the motion dynamics is possible, for instance, through a search for repeatable patterns during the course of the CoP. This can be observed in the frequency domain, for example, using the Fourier’s transform [10] Fast Fourier Transform (FFT) provides information on whether there are any repeatable patterns during the course of CoP measurements and the range of frequencies and amplitudes of the executed movements [1]. On the other hand, Short Time Fourier Transform (STFT) makes it possible to examine changes in the repeatability of some patterns of the CoP displacements in time.

Studies conducted by Liang [11], Błaszczyk [12] and Jurkojć [10] have shown that the observation of specifically defined frequency bands of CoP movements enables determination of the effectiveness of the balance system’s response to balance disruptions, which can be used as an indicator in the diagnosis of the motor system disorders. Research by Davis [13] and Chagdes [1] has shown increased mean power of the spectrum of frequency in the case of tests with closed eyes. A considerable part of the spectrum also shifts in the direction of higher frequencies, which indicates quicker movements around the position of balance [1]. The application of STFT to studies of balance by Michnik [14,15] and Jurkojć [16] has made it possible to define the influence of virtual scenery changes and their parameters on stabilographic values, both in the time domain and the frequency domain. These studies confirm that not only do projection systems affect the way the body balances itself but also the impact of the center projected in virtual reality.

Previous studies have provided a lot of information on the application of frequency analysis to the process of diagnostics of the motor system. However, use of the FFT and STFT methods in the evaluation procedure of the CoP displacements during stabilographic tests has considerable limitations. Measurement devices often takes samples with a frequency higher than 100 Hz (even up to 1000 Hz), making a substantial part of the spectrum impossible to interpret, particularly analysis of energy in bands at low frequencies. The range of frequencies of postural movements and CoP displacements observed in stabilographic tests is within the range of 0 Hz–5 Hz, whereas the spectrum obtained through sampling at a frequency of 1000 Hz is within 0 Hz–500 Hz [1,17]. The oversampling frequency causes the appearance of additional high-frequency noise that does not reflect any actual fluctuation in the postural system. In addition, with an increase in sampling frequency, the useful part of the spectrum for stabilography (0 Hz–5 Hz) is a smaller percentage of the spectrum. The generated noise at high frequencies translates into a decrease in the percentage of energy in the usable bands which affects distortion in the results.

The use of FFT analysis or STFT analysis provides constant resolution on the spectrum side, which is calculated as a ratio of sampling frequency to a given number of samples collected in a certain time interval [10,15]. The analysis uses a small section of the spectrum in which the number of samples depend on the testing time, which is usually 30 s or 60 s. The computed constant value of the spectral resolution does not allow for an accurate interpretation of very low frequencies, for instance around 0.1 Hz. For example when the test lasts 30 s the resolution is 1/30 Hz. In this case, in the frequency range from 0 to 0.1 Hz, we are able to detect a maximum of the 4th harmonic (four peaks in the spectrogram). This energy is expressed by discrete portions and for very low frequencies in the spectrum is not continuous. This indicates the weaknesses of the Fourier transform especially at low frequencies. It is even worse when using the Short-Term Fourier Transform because the time window must be smaller than the signal length and hence the resolution is much lower. Therefore, for example, for the 4 s window used in literature, the resolution is already 0.25 Hz, which completely disqualifies the use of Fourier transform in any analyzes.

A better approach to use in stabilographic analysis is the compression of the observation of frequency bands for low frequencies. This can be achieved using the wavelet transform [1,18]. Direct results of the wavelet transform may be applied to examine coherence values between the motion of a moving room and postural adjustments [19]. In addition, the wavelet decomposition enables the determination of short-term changes in postural stability, which has been shown in previous studies by Singh [20] The studies applied this method to Critical Point Interval as the unit differentiating the stability of the posture between various age groups. Similarly, in the investigations conducted by Nema [21], the wavelet decomposition has been used for the detection of quick changes in stability which may precede, for instance, a fall.

In their studies, Chagdes [1] and Maatar [22] used a wavelet decomposition apparatus and energy calculated in specific bands of the wavelet decomposition to show the existence of changes in postural stability analysis in tests with open and closed eyes. However, the subject literature does not provide any information on such postural stability changes (visible as changes in the frequencies of CoP displacements) occurring as a result of the use of virtual reality projection systems. Due to some of the advantages of the wavelet decomposition connected with a more precise analysis of low frequencies of CoP displacements, this method can be deemed appropriate in stabilographic tests and useful in the analyses of the impact of virtual reality systems on changes in postural stability.

### Target of the Research

The objective of the study was to determine changes in postural stability during tests with open vs. closed eyes and the application of virtual reality projection in the form of a system comprised of a Head Mounted Display device (HMD) and body balance measuring platform. This system is dedicated to the diagnosis of patients with neurological problems, which involve altered mechanisms of body balance. These investigations have been trying to understand how signal energy distribution may on the full frequency spectrum when using virtual reality projection systems such as the HMD system. This study focused both on the visual and mechanical effects of HMD systems with respect to frequency domain and extension to traditional stabilography analyses. The analysis of the energy distribution was conducted using the wavelet decomposition which enables the compression of the observed energy changes in the lower frequency bands of CoP measurements, based on a 30 s study time frame.

## 2. Materials and Methods

### 2.1. Background of Wavelet Analysis

The wavelet transform (WT) is one of the methods of signal analysis in the field of frequency studies. From a mathematical perspective, the WT is the convolution of the time series signal with wavelets of different scales (a) and translations (b) [1,17]. This transformation converts real time series signal *s*(*t*) into a two-dimensional real space with previously mentioned variables *a* and *b*. The result of the calculated wavelet coefficient (WC) at time scale *a* and time instant *b* is shown in dependence (1).
(1)$$WC=\;\int_{-\infty }^{+\infty }s\left(t\right)\psi_{a,b}^{*}\left(t\right)dt,$$
where *s*(*t*) represents time series signal and for purposes of this study, constitutes the CoP course, “*” represents the complex conjugate and *ψ* is a wavelet function-‘child wavelets’ derived from a ‘mother wavelet’ are described in Equation (2).
(2)$$\psi_{a,b}^{\;}(t)\;=\;\frac{1}{\sqrt{a}}\psi_{\;}^{\;}\left(\frac{t-b}{a}\right).$$

The above-mentioned function is a function of time scale “*a*” and is localized near time instant “*b*.” In addition, it must meet various mathematical requirements set by Addison [17] and Morales [23] Mathematical resemblance to the Short Time Fourier Transform (STFT) can be observed here, nevertheless the result obtained by means of the wavelet transform is described in spaces “*a*” and “*b*” and not in space of time and frequency. In the case of discrete wavelet transform (DWT), which is used in digital processing of signals, values “*a*” and “*b*” show the dependence shown in (3).
(3)$$a=\;2^{j\;\;\;\;\;}\Lambda \;\;b=\;2^{jk},$$
where *j* = 1, …, *J* are discrete levels of time scales and *k* = 0…*k*(*j*). The scope of time scales is defined by the range shown in dependence (4).
(4)$$t_{j}=\;\left\{\frac{2^{2^{j-1}}}{f_{s}},\frac{2^{2^{j}}}{f_{s}}\right\},$$
where *f*_*s*_ is signal sampling frequency and *k*(*j*) is expressed in dependence (5).
(5)$$k\left(j\right)\;=\;floor\left(\frac{N-2^{j}}{2^{j}}\right),$$
where floor function rounds down to an integer. Using transformation (3), it can be stated that the wavelet transform in discrete space transforms signal *s*(*t*), which was determined for discrete values *t*, into two-dimensional signal DWT in the space of discrete level *j* and discrete location *k* [1].

At each discrete time level, the DWT may be used for the reconstruction of time series signals at the specified level resulting in a ‘detail’ time series signal-cD(t). Dependence (6). can be applied to such a wavelet decomposition.
(6)$$cD_{j}\left(t\right)\;=\;\overset{K\left(j\right)}{\underset{k=0}{\sum }}DWT\left(j,k\right)\psi_{j,k}\left(t\right).$$

Analogically, dependence (7) can be derived, which constitutes the approximation of the time series signal at specified level *cA* [1,17].
(7)$$cA_{j}\left(t\right)\;=\;\sum_{k=0}^{K\left(j\right)}\mathrm{DWTap}\left(j,k\right)\phi_{j,k}\left(t\right),$$
where ϕ represents the direct scaling function connected with wavelet function ψ, whereas DWTap (*j,k*) represents the DWT approximation defined by dependence (8), analogous to dependence (1).
(8)$$\mathrm{DWTap}\left(j,k\right)=\;\int_{-\infty }^{+\infty }s\left(t\right)\phi_{j,k}\left(t\right).$$

In practice, despite the complex computations dependencies (6) and (7) are implemented in the form of a system of low-pass and high-pass filters. In this way, courses *cD_i_* at a given level are obtained by means of high-pass filtration of base signal *s(t)* or signal *cA_i_*_-_ [1,17,24]. Values *cA_i_* are obtained through the low-pass filtration of signal *cA*_*i*−1_. The whole process is shown in Figure 1 [24].

The limit frequency of filters in the first filtration amounts to a half of the sampling frequency of signal fs and subsequently is decreased by half at each next level. The characteristics of filters depend on the selected wavelet ψ and scaling function ϕ [24]. Signals *cA_i_* and *cD_i_* are expressed as filtered time courses and the whole process constitutes the wavelet decomposition.

### 2.2. Signal Energy Distribution for Wavelet Decomposition

Signal energy distribution *E_Di_*_%_ for wavelet decomposition for *cD_i_* in the time domain can be calculated using formula (9) [1,17].
(9)$$E_{Di\%}=\frac{E_{Di}}{E_{tot}}100\%=\frac{\sum_{t=1}^{T}cD_{i}{}^{2}\left(t\right)\;}{\sum_{t=1}^{T}s^{2}\left(t\right)}\;100\%,$$
where: *E_D_*_i_—energy of *cDi(t)* signal, *E_tot_*—total energy of *s*(*t*) signal, *i*—level of decomposition, *T*—time vector length. The percentage signal energy distribution *E_Di%_* is the same as the percentage power spectral density distribution.

### 2.3. Study Group

The study consisted of 28 participants (14 women and 14 men) with an average age of 22 (standard deviation SD 1.3), average height 173 cm (SD 9) and average weight 66 kg (SD 11.6). None of them had a history of an extreme lower limb injury or suffered from motor system dysfunctions or balance disorders.

### 2.4. Approval

This study was previously approved by the Ethics in Research Committee of the Academy of Physical Education in Katowice (protocol number 11/2015). Each of the study participants gave informed consent in accordance with the Ethics in Research Committee, as well as the form and the course of the study. Consent to participate in the study was confirmed in oral form (written consent was not required).

### 2.5. Measurement Stand

The study was performed using a measurement platform (WinFDM-S, Zebris, sample frequency 100 Hz, 2560 tensometer sensors, sensors area: 34 cm × 5 cm) and Oculus DK2 HMD system. VR application was prepared in Unity3D system and consisted of a simple scenery. The subject’s avatar was placed in a desert, in an empty place near a small castle. A view of the scenery was presented in Figure 2. Distribution of feet pressure on the ground was measured during each test.

### 2.6. Experimental Procedure

The conducted investigations involved measurement of the pressure distribution on the ground. On the basis of pressure distribution and using a software program dedicated to a WinFDM-S platform, the course of displacements of the Center of Pressure (CoP) in the Anterior-Posterior (AP) direction and Mediolateral (ML) direction were determined. The experiments involved three tests: a stabilographic test, a classic Romberg’s test with open eyes (OE) and closed eyes (CE) as well as a test in virtual reality with participants wearing an HMD (VR) system. During the tests, the subject was standing on a stabilographic platform motionlessly with their arms crossed across the chest and hands on the shoulders. Each test lasted at least 50 s. The analysis involved a 30 s course beginning from the 10th second of the examination. The tests were performed in random order.

### 2.7. Analysis of Results

In the first stage of the analysis, the wavelet decomposition of the CoP signal for the ML and AP directions was conducted separately for each subject. The decomposition was performed for 12 levels, similar to Chagdes study [1] and in accordance with the algorithm described in the chapter “Background of wavelet analysis.” As the basic wavelet, the Coiflets wavelet was chosen, specifically Coif5, due to orthogonality and the fact that the wavelet does not affect the energy of signals generated during decomposition [25]. The subsequent step involved the computation of the signal energy distribution for each signal of *cD_i_* and *cA_i_* obtained in the time field, in accordance with the algorithm described in the chapter “Signal energy distribution for wavelet decomposition.” The computations were performed using the author’s algorithm in the Matlab software. The ranges of frequency for signals *cD_i_* and *cA_i_* were determined on the basis of the sampling frequency of data in the system, which amounted to 100 Hz. These ranges are presented in Table 1.

The last phase involved a statistical analysis of the obtained results in Statistica 13.3 software. In the first step, using the Shapiro–Wilk test the researchers tested the occurrence of a normal distribution for the obtained energy values. Due to the unilateral nature of the distributions and skewed results of the Shapiro-Wilk tests, comparisons were made for groups with a distribution divergent from normal. Next, using the Wilcoxon tests with Bonferroni correction for dependent groups, a comparison was carried out for the medians of the energy values obtained for each band of the decomposition.

## 3. Results and Discussion

An example of the CoP signal course for a randomly selected subject and signals *cD_i_* occurring at the subsequent 12 levels of following decomposition as well as the course of signal cA_12_ are presented in Figure 3.

The figure shows the reconstructed waveforms of the wavelet decomposition for individual bands. As the level of decomposition increases (as the number increases), an increase in the amplitudes of the individual decompositions can be observed. For the cD1–cD4 decomposition, the amplitude values are much smaller than for the lower frequencies, which indicates a smaller compensation of energy in these bands calculated in the next steps.

Medians of percentage energy values for the signals occurring in the decomposition process are presented in Figure 4. The above-mentioned medians were calculated for the whole study group within the range of 25%–75%. The values were calculated separately for displacements in the AP and ML planes.

In the first phase of the statistical analyses the researchers examined the occurrence of normal distributions for the calculated values of the percentage distribution of energy for individual signals cD_1_–cD_12_ and cA_12_. All calculations resulted in values *p* < 0.05, which indicated that the distributions diverged from a normal distribution. In the next step, for each set of the calculated energy values in bands cD_1_–cD_12_ a comparison was made between results obtained in the OE, CE and VR tests using the Wilcoxon test with Bonferroni correction.

In the case of courses in the AP plane for bands D_1_–D_4_, the results did not show any statistically significant difference. The first statistically significant differences appeared in band D_5_ where a value of *p* < 0.05 was obtained for all comparisons. In the case of band D_6_, values obtained in the CE test were significantly higher than in the OE and VR tests. In turn, in bands D_7_ and D_8_ values obtained in the CE and VR tests did not differ statistically, whereas the OE test results were statistically lower. In bands D_9_ and D_10_, all energy results did not show any statistically significant differences. Band D_11_ resulted in higher statistical values for the OE test. In bands D_12_ and A_12_, the results did not reveal any statistical differences.

In the case of the courses in the ML plane for bands D_1_–D_5_, the results did not show any statistically significant difference (*p* > 0.05). For bands D_6_ and D_7_, higher values were noted for the CE test than the OE and VR tests. In bands D_8_ and D_9_, no statistically significant differences were observed. In turn, in band D_10_, the values for the OE test were significantly higher than in the case of the CE and VR tests, where no significant differences were noted between these two. Finally, for bands D_11_, D_12_ and A_12_ no significant differences were observed (for all comparisons *p* > 0.05).

## 4. Discussion

The main goal of the presented research was to analyze how posture stability changes as a result of using spatial projection systems in the form of HMD goggles. The use of frequency analysis provides additional information extending standard analyses of balance maintenance. The findings in this study showed that wavelet decomposition made it possible to observe very low frequencies in CoP movement, whereas determination of signal energy enabled observation of percentage distribution of this movement along with individual frequencies.

The obtained profile of the energy distribution (Figure 3) for the OE makes it possible to observe increased values of energy in the bands of lower frequency. Along with the increase in the frequency of CoP motion in Figure 3 for OE for AP direction, the energy values decrease, which is particularly visible in the test in the ML direction. This fact was also confirmed in investigations by Chagdes et al. [1] and analyses using FFT performed by Jurkojć [10]. In the case of tests in the AP direction, a visible increase in the energy for bands D_10_ and D_11_ is an exception. This fact was also confirmed by studies carried out by Chagdes et al. [1] and also observed by Michnik et al. [15]. These bands correspond to the frequency range from 0.024 Hz to 0.098 Hz and the CoP movements in this direction most probably reflect the movements of the thorax while breathing.

The highest energy values for the OE and VR tests were observed in the band of the lowest frequencies. A high value of energy noted for band A_12_, which nears 20% in all tests, points out to the fact that a slow drift (slow motion) occurs in the movements of the CoP displacement. The slow drift energy constitutes one fifth of the total energy. In order to gain more information on the course of the slow drift, it is necessary to prolong the time of performing the tests. It is advisable to conduct diagnostic stabilographic tests in time intervals longer than 30 s, which will enable the examination of low frequencies of the CoP displacements. A recommended time period for the test should be double the one used in this study and should therefore last over 60 s. This will at least double the effective number of samples which are taken into consideration in the process of wavelet decomposition.

A slightly different situation occurs in the scope of high frequencies. In the case of measurements of frequencies higher than 3.125 Hz conducted for the signal in the AP direction and 1.563 Hz for the signal in the ML direction, the results of the OE, CE and VR tests do not show any statistical differences and their values are close to zero. This means that high frequencies do not play a significant role in the process of keeping balance during free standing in healthy persons. For these frequencies, the amplitudes are usually very small, which means that these small fluctuations in different time scales do not contribute to large-scale changes in stability in a manner consistent with the non-linear dynamics of the postural system. Similar conclusions were drawn by Liang et al. [11], in their research, which indicates that the afore-mentioned frequencies are insignificant in the process of fall prediction.

From the perspective of postural stability, the observation of the range of medium frequencies appeared to be the most important in our investigations. The significance of medium frequencies was also confirmed in a stability analysis performed by Iuppariello et al. [25]. In Figure 3, for the D_6_ band in the AP direction and D_6_ and D_7_ bands in the ML direction, there is an increase in energy in the CE test in relation to other tests. In subsequent lower frequency bands, D_7_ and D_8_ in the AP direction as well as D_8_ and D_9_ in the ML direction, this increase is followed by the growth of signal energy for the VR test. In these bands, the energy value in the VR test is at a percentage level comparable to the energy in the CE test, however, it is much higher than in the OE test. In the following bands D_9_, D_10_, D_12_, A_12_ in the AP direction and bands from D_10_ to A_12_ in the ML direction, no statistically significant differences were found, except for band D_11_ where higher values obtained for the AP direction may be related to the previously mentioned breathing process.

What is crucial for the determination of the stability of frequencies in the CoP displacements are observations of the medium frequency bands which showed statistically significant differences in their values. Comparing the obtained values of energy distributions, it can be stated that the VR test values in bands showing statistically significant differences, that is, from D_6_ to D_8_ in the AP direction and D_6_, D_7_, D_10_ in the ML direction, are higher than in the OE test and lower than in the CE test. This may mean that the application of virtual reality in the form of the HMD systems and goggles worsens the postural stability, however, not to such a degree as in the case of stability with closed eyes (CE). The obtained results are closer to the results from the CE test, where the postural stability is lower than in the case of the OE test.

The impact of virtual reality projection systems on postural stability has been shown in previous studies by Robert [9]. The above-mentioned studies revealed an increase in the range of sways and velocity of the CoP movement when a subject is wearing HMD goggles. The increase in the range and velocity is directly connected to the growth of the frequency of the CoP movements in both the AP and ML directions. The conducted experiments confirmed this and provided additional information on the frequency bands in which the energy increase occurs. It was stated that energy values were higher in bands D_7_ and D_8_ in the VR test in relation to the OE test, in both the ML and AP directions. These bands are characterized by frequency ranges of movements of 0.195 Hz to 0.781 Hz, which falls within the range reported by Liang et al. [11]. This range was emphasized as crucial from the perspective of prediction of falls in people with central nervous system issues. The observed increase in energy in this band indicates a higher risk of falling in relation to the OE tests. Similar conclusions were reached in comparing our data with results from other studies, such as a study by Chagdes et al. [1] that involved tests with closed eyes among both young and elderly people. The above-mentioned studies showed that closure of eyes changes the system into a system with an open feedback loop, which results in larger inhalations and an increase in frequency of the acceleration of the center of mass. Detection of frequency sways and the shape of energy distribution diverging from the one obtained in the VR and CE tests may constitute a useful tool for early diagnostics for patients with central nervous system disorders.

The proposed research methodology is also suitable for the diagnosis of balance disorders. Current diagnostic procedures are based on CoP and CoM (Center of Mass) displacements as investigated by Martínez-Ramírez et al. [26]. These results may help determine changes in postural stability similar to those reported by Chagdes et al. [1]. Also, the process of detection of critical point intervals can be used as described by Singh et al. [20]. These results point to the necessity of using a safety harness (a safety device) in the case of therapies involving virtual reality due to the increased risk of falling during the tests.

## 5. Conclusions

Changes in postural stability can be determined by means of virtual reality projection systems on the basis of percent distributions of energy for certain frequency bands of CoP displacements in standing position. Such stability is classified as an intermediate value between the test with open eyes and the test with closed eyes. These results indicate the importance of introducing safety measures such as safety gear during therapy involving virtual reality technology.

The investigations also confirmed the validity of the calculation methods based on the wavelet transform for assessment of CoP displacements in low frequency bands (lower than 0.1 Hz), which is impossible to do with high degree of accuracy when using other calculation methods and the same research designs.

Moreover, the results reveal that 30 s measurements enable the analysis of only 80% of the energy of a given CoP displacements signals. Prolonging the test duration will make it possible to thoroughly analyze lower frequencies and free drifts in postural stability. Longer measurements will enable a more reliable assessment of postural stability with respect to evaluating energy distribution of the tested signal.

## Figures and Tables

**Figure 1 sensors-20-07138-f001:**
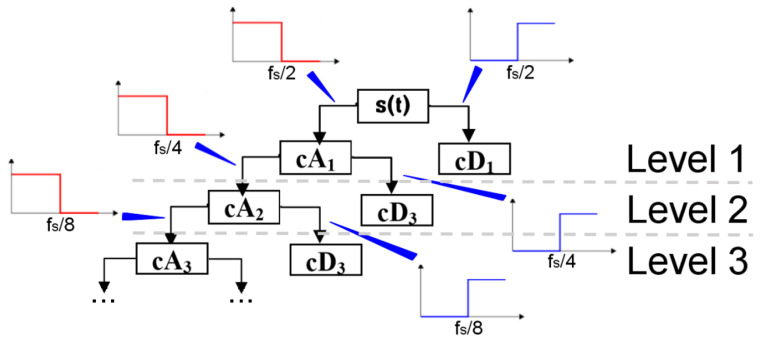
Wavelet decomposition process of s(t) signal on subsequent levels of decomposition.

**Figure 2 sensors-20-07138-f002:**
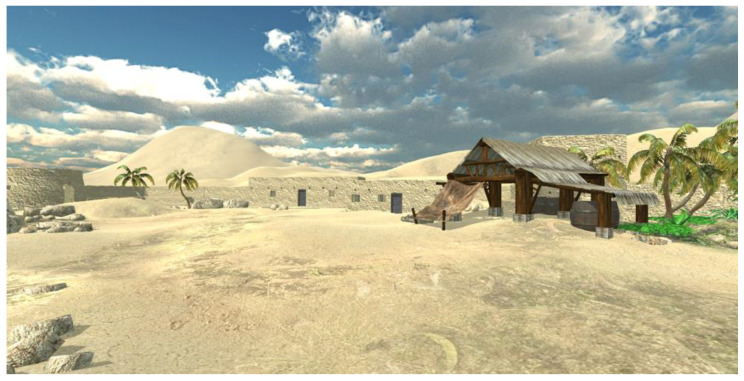
The scenery presented in the study.

**Figure 3 sensors-20-07138-f003:**
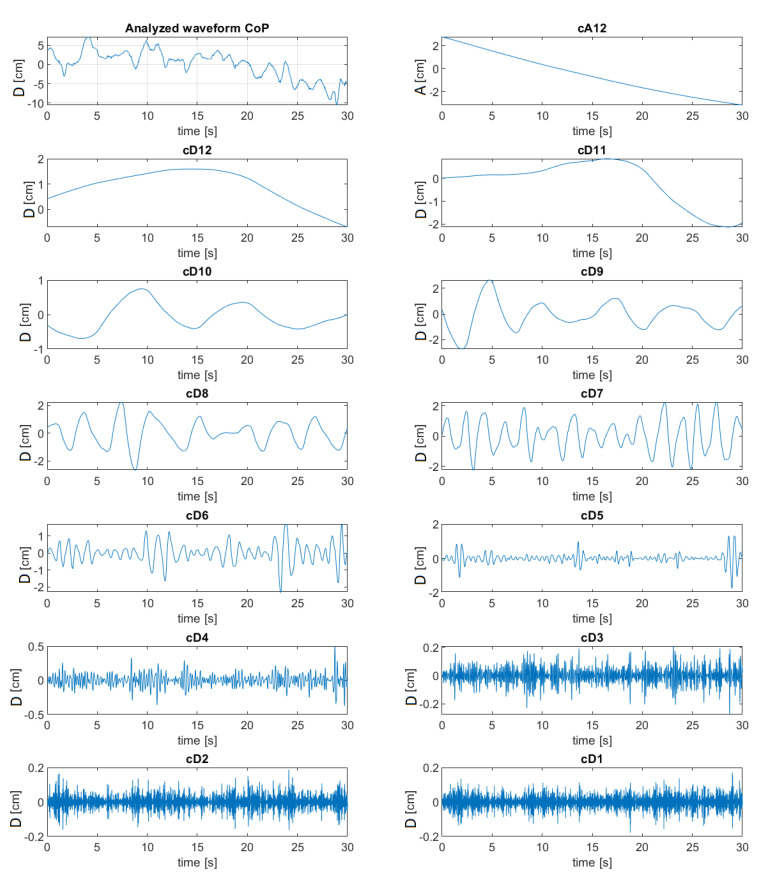
Signals in the field of time at levels subsequent to decomposition of Center of Pressure (CoP) displacement signals for a randomly selected person.

**Figure 4 sensors-20-07138-f004:**
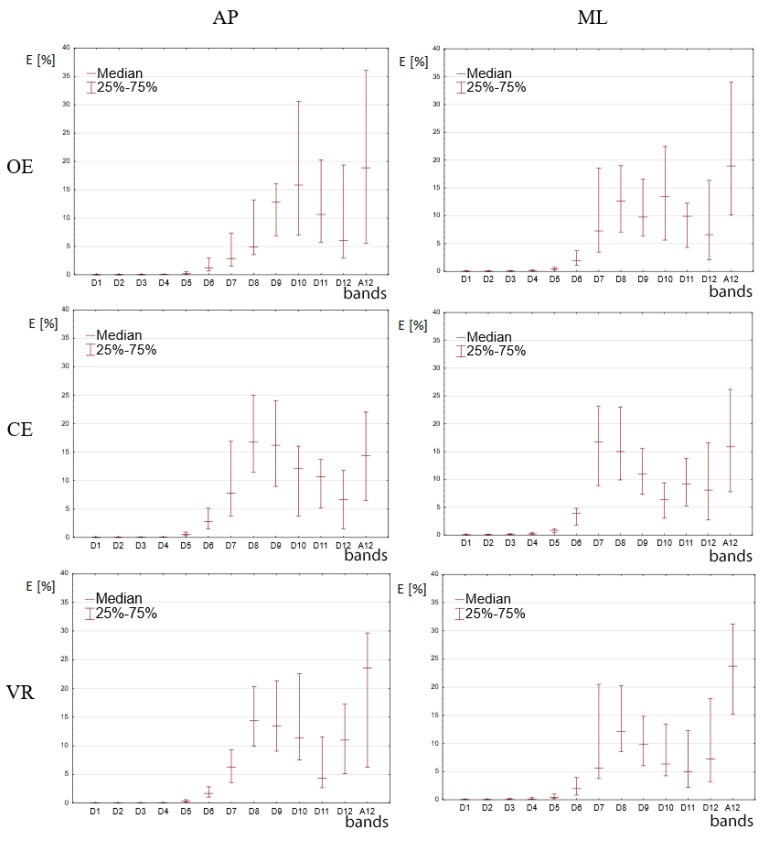
Percentage energy values for individual signals occurring during the decomposition process of the CoP displacement signal in the Anterior-Posterior (AP) and Mediolateral (ML) planes. Results are from the tests with open eyes (OE), closed eyes (CE) and VR (Virtual Reality) projection in the Head Mounted Display (HMD) VR system.

**Table 1 sensors-20-07138-t001:** Frequency bands of signals *cDi* and *cAi* for wavelet decomposition.

Signal from Decomposition	Frequency Band [Hz]	Signal from Decomposition	Frequency Band [Hz]
D_1_	25–50	D_8_	0.195–0.391
D_2_	12.5–25	D_9_	0.098–0.195
D_3_	6.25–12.5	D_10_	0.049–0.098
D_4_	3.125–6.25	D_11_	0.024–0.049
D_5_	1.563–3.125	D_12_	0.012–0.024
D_6_	0.781–1.563	A_12_	0–0.012
D_7_	0.391–0.781

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
