# Peer review of "Wavelet Decomposition in Analysis of Impact of Virtual Reality Head Mounted Display Systems on Postural Stability"

_sensors, 2020, doi:10.3390/s20247138_

Round 1

Reviewer 1 Report

Authors have investigated a problem of body balance in a context of immersive virtual reality interfaces. The problem is valid and worth investigating. Authors have proposed an interesting method exploiting movement frequency bands analysis for the effectiveness of the balance system’s response in three scenarios: one vr-based and two non vr-based, namely VR, CE, OE.  

The literature review section is weakly exhaustive and satisfactory as it reviews related methods mainly qualitatively rather than quantitatively. Statements “better approach” are not supported by any critical analysis of the-state-of-the-art literature. In this context authors’ contribution is vague. Many authors have already attempted to determine changes in postural stability. 

Method’s description is detailed, however formulas are difficult to follow due to very low legibility, but too theoretical and in some aspects weakly embedded in the problem domain. This aspect might be corrected/supplemented. 

Experiments have concentrated on stability characteristics. Displacements of the Centre of Pressure (CoP) in the Anterior-Posterior (AP) direction and Mediolateral (ML) direction were determined. The experiments involved three tests: a stabilographic test, a classic Romberg’s test with open eyes (OE) and closed eyes (CE) as well as a test in virtual reality with participants wearing an HMD (VR) system. 

Unfortunately, there is an incomplete description of the experiment procedure. Much more attention should be put to the stimuli (views changes) which, according to literature, may have a great impact on the reaction of subjects’ equilibrium system. 

In this context research results are probably interesting but difficult to interpret. In consequence experiment procedure (especially stimuli) should be discussed and interpreted in the discussion section. 

From the editorial point of view the manuscript should be thoroughly revised as almost all the formulas are rendered with very low resolution – they are hardly legible. Are parameters “k(j)” and “K(j)” somehow related (line 144 and 147)? The manuscript requires also some minor language corrections. 

Summing up the manuscript hides certain potential but in present form it reveals considerable methodological flaws. It might considered for publication upon completion of mentioned deficits and inconsistencies.  

Author Response

Thank you for careful analysis of our article. The valuable tips have allowed us to improve the article and will be a guide for us to further research in this field. Answering review, we would like to refer to all comments.

  1. “The literature review section is weakly exhaustive and satisfactory as it reviews related methods mainly qualitatively rather than quantitatively. Statements “better approach” are not supported by any critical analysis of the-state-of-the-art literature. In this context authors’ contribution is vague. Many authors have already attempted to determine changes in postural stability.”

Answer:

The authors decided to use the phrase "better" because in the frequency domain it is possible to determine many new indicators such as: CoP change frequency, dominant harmonic disturbance, participation of minor disturbances in the signal and others.

In the frequency domain, it is also possible to determine classical parameters such as: instantaneous and maximum deflections (harmonic amplitudes), scope of CoP motion, average velocity of CoP motion, instantaneous velocity of CoP motion (from the short-term Fourier transform) and others.

More indicators indicate a more modern approach. The authors describe a little more about the comparisons in previous publications:

  1. Jurkojć J, Wodarski P, Bieniek A, Gzik M, Michnik R. Influence of changing frequency and various sceneries on stabilometric parameters and on the effect of adaptation in an immersive 3D virtual environment. Acta Bioeng Biomech [Internet]. 2017;19(3):129–37. Available from: http://www.ncbi.nlm.nih.gov/pubmed/29205224 http://www.actabio.pwr.wroc.pl/Vol19No3/14.pdf
  2. Michnik R, Jurkojć J, Wodarski P, Gzik M, Jochymczyk-Woźniak K, Bieniek A. The influence of frequency of visual disorders on stabilographic parameters. Acta Bioeng Biomech. 2016;18(1):25–33.
  3. Michnik R, Jurkojć J, Wodarski P, Gzik M, Bieniek A. The influence of the scenery and the amplitude of visual disturbances in the virtual reality on the maintaining the balance. Arch Budo. 2014;10(1):133–40. http://archbudo.com/view/abstract/id/10510

The aim of the authors in the Introduction chapter was to outline the necessity to develop methods derived from signal analysis in research on the ability to maintain balance, the possibilities offered by such analyzes and to indicate the purpose of using wavelet analyzes in research in a virtual environment.

The authors are aware of a large number of articles on the research on the ability to maintain balance. The authors' review of balance studies in both real and virtual environments and the use of time and frequency analyzes in such studies includes over 250 items, while being aware that these are not all publications related to this topic. The items cited in the article have been classified by the authors as the most important in terms of the research topics discussed.

The authors decided to carry out the quantitative analysis of the data available in selected literature items in the Discussion chapter.

More specifically, in the last three paragraphs of the discussion chapter, we refer to the results of other works on this subject. Not the quantities but the shapes of energy distributions in wavelet bands were compared. This type of comparison indicates the specification of the dominant energy concentrations in the charts of wavelet bands. It turns out that high gatherings were obtained - as in the works of other authors in the field of eyes open and closed. Research in VR is an innovation and has not been reviewed using such wavelet analysis. The authors did not find other material containing such analyzes for VR using HMD goggles. In addition, the discussion compares the changes that occur in the wavelet bands for closed and open eyes, and it was indicated that the literature provides the same conclusions.

  1. “Method’s description is detailed, however formulas are difficult to follow due to very low legibility, but too theoretical and in some aspects weakly embedded in the problem domain. This aspect might be corrected/supplemented.”

Answer:

The authors agree with the remark, but the poor quality of the equations is a result of the journal's activities. In the previous iteration of the journal article, the authors provided very good quality equations using the equation form in a text document (vector graphics). As a result of editorial activities and the transformation of equations to raster graphics, they lost their quality.

In order to make it easier to read the formulas, the reviewers have prepared a file with the equations only. File attached as additional material

The theory of using wavelet analysis for signal analysis is well known and described. The authors, based on literature analyzes and previous experiences related to the use of FFT and STFT for biomechanical analyzes, decided to extend these analyzes to wavelet analysis for research conducted in a virtual environment. The article describes the methodology and the possibilities of analyzing and interpreting the results of measurements of the ability to maintain balance using wavelets and what changes occur when using a virtual environment (three-dimensional VR images) in relation to measurements with eyes open and closed, which is the aim of the article. An in-depth description of the theory related to wavelet analysis would require additional, multi-page descriptions that would duplicate the content of the cited literature:

  1. Chagdes, J. R.; Rietdyk, S.; Haddad, J. M.; Zelaznik, H. N.; Raman, A.; Rhea, C. K.; Silver, T. A. Multiple Timescales in Postural Dynamics Associated with Vision and a Secondary Task Are Revealed by Wavelet Analysis. Exp. Brain Res. 2009, 197 (3), 297–310. https://doi.org/10.1007/s00221-009-1915-1.
  2. Maatar, D.; Fournier, R.; Lachiri, Z.; Nait-Ali, A. Discrete Wavelet and Modified PCA Decompositions for Postural Stability Analysis in Biometric Applications. J. Biomed. Sci. Eng. 2011, 04 (08), 543–551. https://doi.org/10.4236/jbise.2011.48070.
  3. Nema, S.; Kowalczyk, P.; Loram, I. Wavelet-Frequency Analysis for the Detection of Discontinuities in Switched System Models of Human Balance. Hum. Mov. Sci. 2017, 51, 27–40. https://doi.org/10.1016/j.humov.2016.08.002.

  1. “Unfortunately, there is an incomplete description of the experiment procedure. Much more attention should be put to the stimuli (views changes) which, according to literature, may have a great impact on the reaction of subjects’ equilibrium system.

 In this context research results are probably interesting but difficult to interpret. In consequence experiment procedure (especially stimuli) should be discussed and interpreted in the discussion section.”

Answer:

The authors are aware of the multitude of factors that may affect the values ​​of the quantities describing the ability to maintain balance - both in the real and virtual environment - such as the environment, sounds, well-being, white noise, the accumulation of life problems, etc. In the described research, the authors set themselves the goal of determining how to introduce virtual scenery will change the values ​​of the analyzed quantities for the selected, specific scenery. In this scenery, no additional stimuli that could affect the balance were used - the examined person was presented with the selected scenery in a way that corresponds to being in the real environment - the movement of the head changed the perceived area, which remained stationary with the head still (analogically to the real environment).

The aim of the authors was not to determine how a specific scenery will affect the balance of the examined person, but what changes we will observe for this scenery in relation to the eyes open and closed, and what benefits in such an analysis can be obtained by using wavelets. However, the authors are aware that different types of scenery and the use of additional upsetting stimuli in these scenes may have an impact on the obtained measurement results, as evidenced by previously conducted measurements and analyzes performed both in the time and frequency domains.

  1. Jurkojć J, Wodarski P, Bieniek A, Gzik M, Michnik R. Influence of changing frequency and various sceneries on stabilometric parameters and on the effect of adaptation in an immersive 3D virtual environment. Acta Bioeng Biomech [Internet]. 2017;19(3):129–37. Available from: http://www.ncbi.nlm.nih.gov/pubmed/29205224 http://www.actabio.pwr.wroc.pl/Vol19No3/14.pdf
  2. Michnik R, Jurkojć J, Wodarski P, Gzik M, Jochymczyk-Woźniak K, Bieniek A. The influence of frequency of visual disorders on stabilographic parameters. Acta Bioeng Biomech. 2016;18(1):25–33.
  3. Michnik R, Jurkojć J, Wodarski P, Gzik M, Bieniek A. The influence of the scenery and the amplitude of visual disturbances in the virtual reality on the maintaining the balance. Arch Budo. 2014;10(1):133–40. http://archbudo.com/view/abstract/id/10510
  4. Wodarski Piotr, Jurkojć Jacek, Bieniek Andrzej, Michnik Robert, Gzik Marek, Chrzan Miłosz, Gałązka Mariusz, Herrera liegro Cristina: Evaluation of the changes of gait stability on the treadmill due to treadmill velocity changes by means of time-frequency analysis, Engineering Mechanics 2018, 24rd International Conference, May 14-17, 2018, Svratka, Czech Republic, Editors Cyryl Fisher, Jiri Naprstek, ISBN 978-80-86246-88-8, ISSN 1805-8248, pp. 929-932 https://www.engmech.cz/improc/2018/929.pdf
  5. Wodarski Piotr, Jurkojć Jacek, Gzik Marek, Bieniek Andrzej, Chrzan Miłosz, Michnik Robert: The impact of Virtual Reality on ranges of COP motions during gait, Innovations in Biomedical Engineering, Advances in Inteligent System and Computing, Volume 925, ISSN 2194-5357, s. 218-232, 2019 https://link.springer.com/book/10.1007/978-3-030-15472-1
  6. Wodarski P, Jurkojć J, Polechoński J, Bieniek A, Chrzan M, Michnik R, et al. Assessment of gait stability and preferred walking speed in virtual reality. Acta Bioeng Biomech. 2020;22(1). http://www.actabio.pwr.wroc.pl/Vol22No1/14.pdf

  1. “From the editorial point of view the manuscript should be thoroughly revised as almost all the formulas are rendered with very low resolution – they are hardly legible. Are parameters “k(j)” and “K(j)” somehow related (line 144 and 147)? The manuscript requires also some minor language corrections.”

Answer:

The authors agree with the remark, but the poor quality of the equations is a result of the journal's activities. In the previous iteration of the journal article, the authors provided very good quality equations using the equation form in a text document (vector graphics). As a result of editorial activities and the transformation of equations to raster graphics, they lost their quality.

In order to make it easier to read the formulas, the reviewers have prepared a file with the equations only. File attached as additional material

According to the previous reviewers' suggestions, the article has been proofread by a native speaker of English who holds a PhD in Biomedical Sciences. A proofreading certificate is attached to the additional documents

  1. “Summing up the manuscript hides certain potential but in present form it reveals considerable methodological flaws. It might considered for publication upon completion of mentioned deficits and inconsistencies. “

Answer:

We kindly ask you to consider our revised article ready for publication. The article supplements a series of publications with additional materials and is necessary to continue the discussion on the assessment of postural stability.

Reviewer 2 Report

The authors presented methods to analyze how spatial projection systems influences body balance.

The introduction is informative.

The methods are introduced with details.

The results are discussed properly.

This work might be useful for patients with neurological diseases.

A few suggestions which may further improve the manuscript.

  1. The quality of almost all the equations are very bad. Need to be replaced with proper font size and color (black).
  2. X-Labels are missing for Fig. 4. Resolution of figures can also be improved.
  3. It is not necessary to introduce so many details of wavelet transform, since the knowledge is very general and well-known in the field.
  4. Page 5, line 193, is it typo for the size of sensor area?
  5. All the units would be better if they all changed to standard units. For example, meter instead of centimeter.
  6. The authors may want to discuss more about how signals differ from various subjects, and what are the major reasons for these difference.

Author Response

Thank you for careful analysis of our article. The valuable tips have allowed us to improve the article and will be a guide for us to further research in this field. Answering review, we would like to refer to all comments.

Point 1: The authors agree with the remark, but the poor quality of the equations is a result of the journal's activities. In the previous iteration of the journal article, the authors provided very good quality equations using the equation form in a text document (vector graphics). As a result of editorial activities and the transformation of equations to raster graphics, they lost their quality.

In order to make it easier to read the formulas, the reviewers have prepared a file with the equations only. File attached as additional material

Point 2: A label has been added to the image. All high-resolution images sent to the editorial office. The authors do not know why a low resolution image was inserted into the text in the manuscript. We include high-resolution images in the additional materials

Point 3: The authors decided to present a general course of activities used in the calculations used in the methodology. This was inserted based on suggestions from previous reviewers

Point 4: Corrected

Point 5: As suggested by previous reviewers, the authors used derived units to use as few transformations as possible

Point 6: The case studies will be published as a separate, comprehensive monograph on this subject. We try to present small portions of results in articles, and then an article will be created that will bring the whole thing together.

According to the previous reviewers' suggestions, the article has been proofread by a native speaker of English who holds a PhD in Biomedical Sciences.

Round 2

Reviewer 1 Report

In the revised version of the manuscript Authors have provided a precise and convincing arguments clarifying most of the problems regarding role of wavelet decomposition in a VR HMD systems postural stability analysis. Especially analysis of low frequencies in CoP movements and corresponding signal energies were convincing and supportive.

Nevertheless still some editorial concerns remained. Exemplary still the quality of equations is poor and should be solved before publication. Additionally some inconsistencies in exploited symbols appear, i.e. parameters “k(j)” and “K(j)” mean probably the same but are different (lines 144 and 147), “s” in “fs” (line 147) should be written as a subscript, etc.

Summing up the manuscript is interesting and should be accepted for publication after minor editorial revision.

Author Response

Thank you for your positive review.

Typos have been corrected.

Regarding the low quality of the equations, we are in contact with the editorial office of the magazine and we have sent a special file with equations saved in vector graphics.

Reviewer 2 Report

Line 193 is still wrong.

Author Response

Thank you for your positive review.

Typo have been corrected.

This manuscript is a resubmission of an earlier submission. The following is a list of the peer review reports and author responses from that submission.

Round 1

Reviewer 1 Report

In the manuscript, the authors investigate to determine the changes in postural stability during tests with eyes open and closed, and the application of virtual reality projection in the form of a system with Head Mounted Display device and body balance measuring platform zebris FDM-T. After reading the manuscript with interest, the reviewer found several problems with this paper that require major revisions before being suitable for publication in Sensors journal.

Different questions and remarks.

1. Avoid acronyms in the abstract (VR), line 24.

2. A lot of acronyms (3D, CoP, SD...) are used in the paper and never explain,

3. Line 45 Romberg’s test should be explained and reference should be added for a non-specialist to understand.

4. Paragraph 2.1. The background of wavelet analysis is too long (too many equations not very useful for this paper).

5. Paragraph 2.5. the Gait analysis FDM-T measurement platform is not described and add a figure to present the VR scenery.

6. Paragraph 3. Results and discussion and Paragraph 4. is also Discussion.

7. 8 references for this paper are self-citations by authors. Are all these self-citations justifying?

Author Response

Thank you for careful analysis of our article. The valuable tips have allowed us to improve the article and will be a guide for us to further research in this field. Answering review, we would like to refer to all comments.

Point 1: The abbreviation was expanded and the abstract was shortened to 200 words

Point 2: Appropriate corrections have been added to the text. Acronyms are explained in parentheses.

Point 3: An explanation was added

Point 4: The background of the wavelet analysis has been added as a pre-chapter of the research methodology. This chapter shows in a mathematical manner the use of wavelet analysis in the conducted research. This analysis is very extensive and has many modifiable variables, hence it was concluded that a detailed presentation of the calculation method will allow the methodology to be recreated and to be used for other studies.

Point 5: The FDM-S stabilographic platform was used in the research, FDM-T gait analysis was not carried out, an error appeared in the text, which was corrected. More detailed information on the platform has been added to the text.

The appearance of the scenery has not been added to the article as we believe that it doesn't change much. There is, a brief description of what the application looks like. However, the assessment of changes in stability was carried out comprehensively, taking into account the influence of all factors: goggle weight, image refresh rate, image width, immersion level and scene appearance. If the reviewers, after a short comment, continue to maintain the necessity to add a scene as a necessary element - the scene will be added.

Point 6: The appearance of the subsection discussion separates the presentation of the results from the actual discussion.

Point 7: The publication is a continuity of the presented research. In the first stage of performing frequency analysis using FFT (Fast Fourier Transform). Then, the stabilographic tests were analyzed by the authors using the  time-frequency analysis STFT (Short-Term Fourier Transform). In the next steps in previous publications, the advantages and disadvantages of the analyzes were presented. Additionally, previous studies analyzed the effect of the type of scenery and the movements of the scenery on human motor skills and the ability to maintain balance. Currently, research methods have been enriched with wavelet analysis. The continuity of the research conducted leads to self-citations, which at the same time build a history of activities and allow for the analysis of the authors' earlier works in the already continued 15 years of research tradition.

According to the reviewers' suggestions, the article has been proofread by a native speaker of English who holds a PhD in Biomedical Sciences.

Reviewer 2 Report

This work analyses how the content of head-mounted displays affects the users' postural stability. The authors carried out the systematic analysis and invited 28 subjects to do the human factor experiments. Through their analysis and the experiment, this work gives a guideline to indicate the importance of using safety measures while users immerse themselves into the virtual reality. This manuscript also discusses the assessment of CoP using the wavelet transform method. It finally concludes that longer measurements lead to a more reliable assessment in the measurement tests.

The following comments are for the authors to improve their work:

  • Line 146, what does a ‘detail’ time series signal mean? Please clarify it. 
  • Figure 2, the profiles show the result of a randomly selected person. Please show one or two more randomly selected person/people inside the figures (use different colours or line styles to indicate different people). In the way, we can get a general trend among the selected people.
  • Please briefly discuss each profile in Figure 2.
  • Section 2.3, What does “SD” stand for? (standard deviation?)
  • Table 1, what do the “12,5”, “6,25”, “3,125”, etc mean? Are you trying to say “12.5”, “6.25”, “3.125”, etc? If so, please correct those.
  • There are several typos and grammatical errors in the manuscript. Please correct those.
    • Line 26: frequencies of the “the” centre of the pressure signal
    • Line 79: The use of FFT analysis or STFT analysis “obtains” the constant resolution
    • Line 86: four peaks in “spectrogram”
    • Line 259: distribution of this movement along with individual frequencies. 
    • There are many articles missed in the manuscript, such as “the” band of low frequency, “the” normal distribution, etc.

Author Response

Thank you for careful analysis of our article. The valuable tips have allowed us to improve the article and will be a guide for us to further research in this field. Answering review, we would like to refer to all comments.

Point 1: Wavelet coefficients at each of the discrete levels of timescales can be thought of as ‘detail Wavelet coefficients’ and can be used in the reconstruction of the time series signal at the specified level resulting in a ‘detail’ time series signal.

‘detail’ time series signal - means a reconstructed signal with the use of wavelet coefficients at a specific level of decomposition. In this context, the term has been used in:
Addison P (2002) The illustrated wavelet transform handbook: introductory theory and applications in science, engineering, medicine and finance. CRC Press, New York

Point 2: While for cD8 - cD12 it would be possible to distinguish the waveforms of different people, the remaining graphs should be presented in separate figures. The resulting related graphs would be related. Adding more pictures would confuse the text. The authors showed an example to demonstrate what the results obtained as a result of the wavelet distribution looked like and to show for which waveforms the energies were calculated and then averaged.

Point 3: A brief commentary on the waveforms was introduced in the text

Point 4: SD means standard deviation. Appropriate corrections have been added to the text

Point 5: Corrected

Point 6: Typos and grammatical errors were corrected in text

According to the reviewers' suggestions, the article has been proofread by a native speaker of English who holds a PhD in Biomedical Sciences.

Reviewer 3 Report

In this study, the authors experimented with the stability of wearing a VR-HMD while standing upright compared to a condition without an HMD. They obtained data on the movements of the center of pressure (CoP). They used the wavelet transform in their evaluation. The wavelet transform enables us to investigate the low-frequency components.

This is an important topic for the safety of VR-HMDs because of their widespread use. It also has the potential to be useful for testing the keeping balance abilities of older adults.

However, I think that there are not enough experiments to evaluate stability. In addition, the paper lacks necessary information for the readers. Therefore, my decision is made to reject it.

First of all, only one image was used in the experiment as the image of the VR environment. However, the variability of the CoP is likely to change depending on the image. For example, when a user watches a swirling image or an image with a strong brightness change, they are likely to be very wobbly. If the authors want to properly investigate how spatial projection systems affect the movements of the CoPs, they should evaluate them on multiple videos.
As the effects of multiple images are clarified, the information can be provided in the guidelines.

There is little information about the images used in the experiment in the paper. Why does the image include the desert scene? Why is there a small castle nearby? Is the image a video? Or is it a still image? Screenshots of the image you showed should be included in the paper.

There is also the effect of just wearing HMDs. It is necessary to include in the experimental conditions that the eye is closed while wearing the HMD.

How much does the Oculus DK2 weigh? I think the weight of the VR-HMD will also affect the results. If possible, it would be good to have the position of the center of gravity of the HMD.

It is written in the third paragraph of Section 1 that CoP stands for Center of pressure, but please write it in the second paragraph.

Below are just comments.

In this paper, the authors compared OE, CE, and VR. The graph in Figure 3 should be modified to make it easier to compare the results. Or the authors should add other graphs of the results.

Although this is a paper on wavelet transformations, it would be more helpful to add the following items obtained in the present experiment: the range of postural sways of the CoP, mean velocity of the CoP, and maximum velocity of the CoP.

The figures are hard to see. The figures should be a vector image, not a raster image.

Author Response

Thank you for careful analysis of our article. The valuable tips have allowed us to improve the article and will be a guide for us to further research in this field. Answering review, we would like to refer to comments.

The aim of the research was not to investigate all dependencies and types of scenery and their impact on the balance system, but only to determine whether this type of impact occurs and whether it can be detected using wavelet analysis. The authors are fully aware of the fact that the construction of the virtual reality scene and the type of projection conducted differently affect the subjects. We have several scientific publications describing this impact in fully immersive systems such as the cave 3D . The usefulness of various research methods on the impact of VR on body balance has been confirmed and verified on the basis of these previously conducted studies.

This effect has been presented in the following publications:

  1. Jurkojć J, Wodarski P, Bieniek A, Gzik M, Michnik R. Influence of changing frequency and various sceneries on stabilometric parameters and on the effect of adaptation in an immersive 3D virtual environment. Acta Bioeng Biomech [Internet]. 2017;19(3):129–37. Available from: http://www.ncbi.nlm.nih.gov/pubmed/29205224 http://www.actabio.pwr.wroc.pl/Vol19No3/14.pdf
  2. Michnik R, Jurkojć J, Wodarski P, Gzik M, Jochymczyk-Woźniak K, Bieniek A. The influence of frequency of visual disorders on stabilographic parameters. Acta Bioeng Biomech. 2016;18(1):25–33.
  3. Michnik R, Jurkojć J, Wodarski P, Gzik M, Bieniek A. The influence of the scenery and the amplitude of visual disturbances in the virtual reality on the maintaining the balance. Arch Budo. 2014;10(1):133–40. http://archbudo.com/view/abstract/id/10510
  4. Wodarski Piotr, Jurkojć Jacek, Bieniek Andrzej, Michnik Robert, Gzik Marek, Chrzan Miłosz, Gałązka Mariusz, Herrera liegro Cristina: Evaluation of the changes of gait stability on the treadmill due to treadmill velocity changes by means of time-frequency analysis, Engineering Mechanics 2018, 24rd International Conference, May 14-17, 2018, Svratka, Czech Republic, Editors Cyryl Fisher, Jiri Naprstek, ISBN 978-80-86246-88-8, ISSN 1805-8248, pp. 929-932 https://www.engmech.cz/improc/2018/929.pdf
  5. Wodarski Piotr, Jurkojć Jacek, Gzik Marek, Bieniek Andrzej, Chrzan Miłosz, Michnik Robert: The impact of Virtual Reality on ranges of COP motions during gait, Innovations in Biomedical Engineering, Advances in Inteligent System and Computing, Volume 925, ISSN 2194-5357, s. 218-232, 2019 https://link.springer.com/book/10.1007/978-3-030-15472-1
  6. Wodarski P, Jurkojć J, Polechoński J, Bieniek A, Chrzan M, Michnik R, et al. Assessment of gait stability and preferred walking speed in virtual reality. Acta Bioeng Biomech. 2020;22(1). http://www.actabio.pwr.wroc.pl/Vol22No1/14.pdf

In the presented research above, the HMD system in the form of goggles was used. The influence of this system on the ability to maintain balance has been treated as a whole - despite the awareness that the weight of the device, type of projection, display frequency, refreshing and viewing angle all have an influence on stability.

In the conducted research, we focus on showing the usefulness of wavelet analysis in detecting changes in the stability of posture in two different environments. The presented methodology is predisposed to clinical activities.

Of course, this tool will be used to further analyzes of the impact of individual factors on the body balance. This type of research, as suggested by the reviewer, will require the provision of equal conditions for the impact of other factors. There may be an infinite number of scenarios used in research, but they need to be limited and adapted to, for example, the diagnosis or rehabilitation of specific diseases and patients' abilities.

Minor: The results were exported from Statistica Software and for easy comparison, each drawing was scaled the same. Graphs are grouped and placed side by side for visual comparison.

According to the reviewers' suggestions, the article has been proofread by a native speaker of English who holds a PhD in Biomedical Sciences.

Round 2

Reviewer 1 Report

Modifications have been carried out by the authors, but the reviewer still found problems with this paper that require revisions before being suitable for publication in Sensors journal.

Information (photos or video) about the VR scenery are still missing, and are necessary for the reproducibility and understanding of this study.

How did the authors choose their content? Did they try with different sceneries?

Reviewer 3 Report

I understand the aim of this study. However, if the study is only for such aim, I don't know what the novelty and usefulness are.

- novelty
The effect of wearing HMDs seems to be known from previous studies.
We know that wavelet analysis can be used to analyze low-frequency components.
What is the novelty of this study?

- usefulness
It is not clear whether the results obtained can be applied to clinical activities or to the diagnosis of patients with neurological problems.
The authors have not collected data on people with some problems. Their analysis method does not necessarily provide suitable results for such people.

Even if the answers to the above questions are satisfactory, I believe that experiments under closed-eye conditions with HMDs are necessary. We do not know whether the effects are caused by the images on the HMDs or just by wearing the HMDs. Whether this difference can be analyzed by the wavelet method is not clear from the present results.

Information about the HMD scene is necessary for reproducibility.
Is it a still image or a video?
What does the image look like as seen by the participants (screenshot)?
Why did the authors choose those images?